# On-Site Calibration of an Electric Drive: A Case Study Using a Multiphase System

**DOI:** 10.3390/s23177317

**Published:** 2023-08-22

**Authors:** David Soto-Marchena, Federico Barrero, Francisco Colodro, Manuel R. Arahal, Jose L. Mora

**Affiliations:** 1Department of Electronic Engineering, University of Seville, 41092 Seville, Spainjolumo@us.es (J.L.M.); 2Department of System Engineering and Control, University of Seville, 41092 Seville, Spain; arahal@us.es

**Keywords:** electrical drive performance, current sensors, speed measurement, calibration methods

## Abstract

Modern electric machines are attracting great interest from the research community as a result of the increasing number of current applications, including electric vehicles and wind power generators, among others. Different machines, power converters, and control technologies are used, and the number of sensors is usually minimized to reduce the total cost of the system. Particularly interesting are the current and speed sensors, which are essential to the normal functioning of the entire system. This work analyzes different calibration techniques of these sensors, using as a case example a five-phase induction motor drive. Experimental results are included to show the influence of calibration techniques on the system analyzed. The obtained results can be extrapolated to any other similar system.

## 1. Introduction

Electric drives play an important role in modern society and are the mainstay of many industrial applications, such as electric vehicles, elevators, and robotics [1]. They are systems based on electrical machines and controlled by power electronic converters. Their goal is to provide precise transmission between electrical energy and mechanical motion with fast dynamic responses [2]. Among modern electric drives, those with multiphase machines have become popular in the last few decades [3]. Their most important feature is their inherent redundancy, which allows a certain degree of fault tolerance without the use of additional hardware [4]. Moreover, their complexity has recently attracted a large amount of scientific work on the development of controllers that share a common goal: the implementation of an electrical variator with the best current and torque/speed performances [5,6].

Sensors are required to monitor the operating state of the electric drive and feed it back as input to the controller [7]. Continuous monitoring and observation are essential for high-quality controlled systems, which require reliable sensors. The number and type of mechanical and electrical sensors required are variable in electric drives and depend on multiple characteristics of the system [8,9]. In any case, sensor reduction is a critical obligation in most industry applications to reduce the cost and maintenance of the entire system while maintaining its reliability [10,11,12]. This is particularly interesting in the field of multiphase drives, where, for example, the number of current sensors depends on the number of phases. If the neutral point of an electric machine with n phases is isolated, the sum of all phase currents is null, and the number of current sensors required is *n* − 1.

Most researchers and industry applications calibrate the sensors required to ensure accurate measurements in testing systems [13]. This has been the norm since the first application of multiphase systems in the late 1960s, where some advantages over conventional three-phase systems were highlighted [14]. To the best of our knowledge, there is little scientific evidence of the influence of the calibration technique on the measurement of stator currents, where, for example, the performance of the untested phase current, the influence of the calibration technique on the measurement of the rotor speed, and the performance of the closed-loop drive have barely been studied [15,16,17]. Our work analyzes these influences, establishing a reliable procedure for the calibration of the analyzed system that can be extrapolated to any type of electrical drive controlled in speed and current.

The paper is organized as follows. Section 2 summarizes the five-phase induction motor model, which is used as a case study in our research. The calibration procedures analyzed are presented in Section 3, and their accuracies are studied experimentally in Section 4. The conclusions are provided in the Section 5.

## 2. Basics of the Case Study

In general, the symmetrical five-phase induction machine (5pIM), fed by a two-level voltage source inverter (2LVSI) with isolated neutral and distributed windings, is one of the most popular in industrial applications and the research community; see Figure 1 and Figure 2. The 2LVSI generates 32 voltage vectors (30 active and 2 zero-voltage vectors) characterized by the switching vector [*S_a_ S_b_ S_c_ S_d_ S_e_*] as follows:(1)$$\left[\begin{array}{c} v_{sa}\\ v_{sb}\\ v_{sc}\\ v_{sd}\\ v_{se} \end{array}\right]=\frac{V_{dc}}{5}\cdot \left[\begin{array}{c} 4\\ -1\\ -1\\ -1\\ -1 \end{array}\begin{array}{c} -1\\ 4\\ -1\\ -1\\ -1 \end{array}\begin{array}{c} -1\\ -1\\ 4\\ -1\\ -1 \end{array}\begin{array}{c} -1\\ -1\\ -1\\ 4\\ -1 \end{array}\begin{array}{c} -1\\ -1\\ -1\\ -1\\ 4 \end{array}\right]\cdot \left[\begin{array}{c} S_{a}\\ S_{b}\\ S_{c}\\ S_{d}\\ S_{e} \end{array}\right],$$
where the midpoint of the external power supply (*V_dc_*) is considered the ground of the electrical system, and a balanced load is assumed, which is *v_sa_* + *v_sb_* + *v_sc_* + *v_sd_* + *v_se_* = 0 and *i_sa_* + *i_sb_* + *i_sc_* + *i_sd_* + *i_se_* = 0.

The electromagnetic performance of the machine is then represented with a set of phase voltage equilibrium equations of the stator and the rotor in the stationary reference frame (*a*, *b*, *c*, *d*, *e*), although it is usually represented for clarity and simplicity with the Clarke decoupled model using two orthogonal planes *α* – *β* and *x* – *y* plus the homopolar component *z*. This groups the fundamental frequency together with harmonics of order 10 · *k* ± 1 (*k* = 0, 1, 2, etc.) in the *α* – *β* plane, the harmonics of order 10 · *k* ± 3 in the plane *x* – *y*, and project harmonics of order 5 · *k* on the *z* axis. A Clarke transformation matrix must be used to multiply the voltage/flux/current vectors of the stator and rotor in the frame (*a*, *b*, *c*, *d*, *e*), leading to an invariant transformation of magnitudes in the (*α*, *β*, *x*, *y*, *z*) frame. The detailed machine model in the reference frame (*a*, *b*, *c*, *d*, *e*) is detailed in Equations (2)–(17), while the (*α*, *β*, *x*, *y*, *z*) frame can be found in [18].
(2)$$\begin{array}{c} \left[v_{s}\right]=\left[R_{s}\right]\cdot \left[i_{s}\right]+\frac{d}{\mathrm{dt}}\left[\psi_{s}\right]=\left[R_{s}\right]\cdot \left[i_{s}\right]+\left[L_{\mathrm{ss}}\right]\frac{d}{\mathrm{dt}}\left[i_{s}\right]+\frac{d}{\mathrm{dt}}\left[L_{\mathrm{sr}}\left(\theta \right)\right]\cdot \left[i_{r}\right] \end{array},$$
(3)$$\begin{array}{c} \left[v_{r}\right]=\left[R_{r}\right]\cdot \left[i_{r}\right]+\frac{d}{\mathrm{dt}}\left[\psi_{r}\right]=\left[R_{r}\right]\cdot \left[i_{r}\right]+\left[L_{\mathrm{rr}}\right]\frac{d}{\mathrm{dt}}\left[i_{r}\right]+\frac{d}{\mathrm{dt}}\left[L_{\mathrm{rs}}\left(\theta \right)\right]\cdot \left[i_{s}\right] \end{array},$$
(4)$$\left[v_{s}\right]={\left[\begin{array}{ccccc} v_{\mathrm{sa}} & v_{\mathrm{sb}} & v_{\mathrm{sc}} & v_{\mathrm{sd}} & v_{\mathrm{se}} \end{array}\right]}^{T}$$
(5)$$\left[v_{r}\right]={\left[\begin{array}{ccccc} 0 & 0 & 0 & 0 & 0 \end{array}\right]}^{T}$$
(6)$$\left[i_{s}\right]={\left[\begin{array}{ccccc} i_{\mathrm{sa}} & i_{\mathrm{sb}} & i_{\mathrm{sc}} & i_{\mathrm{sd}} & i_{\mathrm{se}} \end{array}\right]}^{T}$$
(7)$$\left[i_{r}\right]={\left[\begin{array}{ccccc} i_{\mathrm{ra}} & i_{\mathrm{rb}} & i_{\mathrm{rc}} & i_{\mathrm{rd}} & i_{\mathrm{re}} \end{array}\right]}^{T}$$
(8)$$\left[\psi_{s}\right]={\left[\begin{array}{ccccc} \psi_{\mathrm{sa}} & \psi_{\mathrm{sb}} & \psi_{\mathrm{sc}} & \psi_{\mathrm{sd}} & \psi_{\mathrm{se}} \end{array}\right]}^{T}$$
(9)$$\left[\psi_{r}\right]={\left[\begin{array}{ccccc} \psi_{\mathrm{ra}} & \psi_{\mathrm{rb}} & \psi_{\mathrm{rc}} & \psi_{\mathrm{rd}} & \psi_{\mathrm{re}} \end{array}\right]}^{T}$$
(10)$$\left[R_{s}\right]=R_{s}\cdot \left[I_{5}\right]$$
(11)$$\left[R_{r}\right]=R_{r}\cdot \left[I_{5}\right]$$
(12)$$\left[L_{\mathrm{ss}}\right]=L_{\mathrm{ls}}\cdot \left[I_{5}\right]+M\cdot \left[\Lambda \left(\vartheta \right)\right]$$
(13)$$\left[L_{\mathrm{rr}}\right]=L_{\mathrm{lr}}\cdot \left[I_{5}\right]+M\cdot \left[\Lambda \left(\vartheta \right)\right]$$
(14)$$\left[L_{\mathrm{sr}}\left(\theta \right)\right]={\left[L_{\mathrm{rs}}\left(\theta \right)\right]}^{T}=M\cdot \left[\Delta \left(\theta \right)\right]$$
(15)$$\theta =\int_{0}^{t}\omega_{r}\mathrm{dt}$$
(16)$$\left[\Lambda \left(\vartheta \right)\right]=\left[\begin{array}{ccccc} 1 & \cos\left(\vartheta \right) & \cos\left(2\vartheta \right) & \cos\left(3\vartheta \right) & \cos\left(4\vartheta \right)\\ \cos\left(4\vartheta \right) & 1 & \cos\left(\vartheta \right) & \cos\left(2\vartheta \right) & \cos\left(3\vartheta \right)\\ \cos\left(3\vartheta \right) & \cos\left(4\vartheta \right) & 1 & \cos\left(\vartheta \right) & \cos\left(2\vartheta \right)\\ \cos\left(2\vartheta \right) & \cos\left(3\vartheta \right) & \cos\left(4\vartheta \right) & 1 & \cos\left(\vartheta \right)\\ \cos\left(\vartheta \right) & \cos\left(2\vartheta \right) & \cos\left(3\vartheta \right) & \cos\left(4\vartheta \right) & 1 \end{array}\right]$$
(17)$$\left[\Delta \left(\theta \right)\right]=\left[\begin{array}{ccccc} \cos\left(\Delta_{1}\right) & \cos\left(\Delta_{2}\right) & \cos\left(\Delta_{3}\right) & \cos\left(\Delta_{4}\right) & \cos\left(\Delta_{5}\right)\\ \cos\left(\Delta_{5}\right) & \cos\left(\Delta_{1}\right) & \cos\left(\Delta_{2}\right) & \cos\left(\Delta_{3}\right) & \cos\left(\Delta_{4}\right)\\ \cos\left(\Delta_{4}\right) & \cos\left(\Delta_{5}\right) & \cos\left(\Delta_{1}\right) & \cos\left(\Delta_{2}\right) & \cos\left(\Delta_{3}\right)\\ \cos\left(\Delta_{3}\right) & \cos\left(\Delta_{4}\right) & \cos\left(\Delta_{5}\right) & \cos\left(\Delta_{1}\right) & \cos\left(\Delta_{2}\right)\\ \cos\left(\Delta_{2}\right) & \cos\left(\Delta_{3}\right) & \cos\left(\Delta_{4}\right) & \cos\left(\Delta_{5}\right) & \cos\left(\Delta_{1}\right) \end{array}\right]$$
where *v*, *i*, and *ψ* denote the voltage, current, and flux linkage, respectively, and the indices *s* and *r* represent the stator and rotor variables, respectively. The subscripts *a*, *b*, *c*, *d* and *e* identify the phases considered, the superscript *T* designates the transpose operator, and *ω_r_* is the electrical speed of the rotor. [*I*_5_] is the identity matrix of order 5; *R_s_* and *R_r_* are the resistance of the stator and the rotor, respectively; *M* is the mutual inductance parameter; and *L_ls_* and *L_lr_* are the leakage inductance parameters of the stator and the rotor, respectively. Finally, ∆*_k_* values are angles defined as ∆*_kv_* = *θ* + (*k* − 1)ϑ, with *k* = {1, 2, 3, 4, 5}, where *θ* represents the instantaneous azimuth of the rotor with respect to the *α*-axis of the stationary reference frame.

This electrical model is complemented with the mechanical equation of the drive:(18)$$J_{m}\frac{d\omega_{m}}{dt}+B_{m}\omega_{m}=T_{e}-T_{L}$$
where $\omega_{m}$ is the mechanical speed of the rotor shaft ($\omega_{r}=p\cdot \omega_{m}$, where *p* is the number of pole pairs), $T_{L}$ is the load torque applied to the machine, $T_{e}$ is the electromagnetic torque, $J_{m}$ is the inertia of the rotating masses, and $B_{m}$ is the friction coefficient. The electromagnetic torque is responsible for the conversion of electromechanical energy that links the electrical and mechanical subsystems. This torque is obtained with the following equation:(19)$$T_{e}=T_{e}=\frac{P}{2}{\left[i_{s}\right]}^{T}\frac{d[L_{sr}\left(\theta \right)]}{d\theta }[i_{r}]$$
which considers that the stator and rotor inductance matrices do not depend on the rotor position and shows that the electromagnetic torque is created entirely from the interaction between the stator and the rotor.

## 3. Calibration Procedures

The calibration process is applied to a laboratory setup based on 5pIM, a five-phase 2LVSI constructed from two SEMIKRON SKS 22F modules, and a DC power supply acting as a DC link (KDC 300-50 from California Instruments Corporation). A diagram of the laboratory setup is shown in Figure 3, where some photographs of the hardware used are also provided, and the main electrical and mechanical parameters of the system are given in Table 1. The control unit is based on an MSK28335 board with a TMS320F28335 digital signal processor model from Texas Instruments. The feedback of the mechanical speed signal is provided by the GHM510296R/2500 quadrature encoder, while four LH25-NP Hall effect sensors from LEM are used to measure the stator phase current signals. A DC motor is also used to generate a torque load (*T_L_*).

### 3.1. Calibration of Current Sensors

The calibration test of the current sensors is based on DC excitation using the power supply, where the electrical machine is driven to have a standstill performance in the stationary state. Therefore, the rotation of the drive is avoided, reducing any influence of the electromechanical system on the procedure. One stator phase of the machine is supplied with a DC voltage step, while the rest of the phases are ground-connected; see Figure 4 for a better understanding of the testing scheme. Forced stator currents are measured using the microcontroller system (MSK28335 board and LH25-NP sensors) and compared with measured stator currents using a MDO4024C scope (characterized by an 8-bit resolution with a bandwidth of 200 MHz) and TCP0020 current probes. The electrical performance of the system is then represented by the following equation:(20)$$V_{s}=\left(R_{s}+\frac{R_{s}}{4}\right)i_{s}$$

The calibration technique applied in most commercial and research five-phase electric drives considers every measured phase independently and is performed at a standstill. Then, four lines of the form $i_{k}=g_{k}s_{k}-o_{k}$ are found, where $i_{k}$ is the measured stator phase current and $s_{k}$ is the 12-bit digital value found in the DSP after an analog-to-digital conversion. The gain and offset values, $g_{k}$ and $o_{k}$, respectively, are obtained after using a least mean squares algorithm between the measured stator currents using the MDO4024C scope and the stator current value obtained by the DSP, $i_{k}$. Figure 5 and Figure 6 summarize the results obtained, where the correspondence between the values obtained using the scope and the fitted lines in the DSP can be appreciated.

Note that the measured stator currents correspond to the stator phases *a*, *b*, *d*, and *e*, where LH25-NP Hall current sensors are used. Also note that the values used to fit the lines in the linear regression analysis are the average value obtained from the MDO4024C data in a calibration test. A generalized alternative to the previous calibration method has also been considered, where a system of equations is introduced instead of designing four independent best-fit curves. Four of the equations are in the form of the best fit curves, $i_{f}=g_{f}s_{f}-o_{f}$, while an extra equation for the unmeasured stator phase current is added. This additional equation is based on the Kirchhoff current law, where the measured stator currents are substituted with their best-fit curves. Then, the following set of equations is obtained:(21)$$\left[\begin{array}{cccccccc} s_{a,1} & -1 & 0 & 0 & 0 & 0 & 0 & 0\\ 0 & 0 & s_{b,2} & -1 & 0 & 0 & 0 & 0\\ -s_{a,3} & 1 & -s_{b,3} & 1 & -s_{d,3} & 1 & -s_{e,3} & 1\\ 0 & 0 & 0 & 0 & s_{d,4} & -1 & 0 & 0\\ 0 & 0 & 0 & 0 & 0 & 0 & s_{e,5} & -1 \end{array}\right]\left[\begin{array}{c} g_{a}\\ o_{a}\\ g_{b}\\ o_{b}\\ g_{d}\\ o_{d}\\ g_{e}\\ o_{e} \end{array}\right]=\left[\begin{array}{c} i_{a}\\ i_{b}\\ i_{c}\\ i_{d}\\ i_{e} \end{array}\right]$$
where the best fit curves are obtained again using a least mean squares method. Consequently, the importance of the errors of any given phase is directly proportional to the number of experiments considered for that phase. This could be interesting because a better performance for the non-measured stator current is expected at the expense of the performance of the other phases. The obtained results are summarized in Figure 7 and Figure 8 and produce quite similar fitting curves to the previous calibration technique using a much more complex calibration procedure.

### 3.2. Calibration of the Speed Sensor

Mechanical speed is measured using the eQEP (enhanced quadrature encoder pulse) module of the DSP. This DSP peripheral is used for direct interface with the incremental encoder to obtain the direction and speed information of the rotating machine from the alternating pattern of dark and light lines (pulses) that occur in the encoder per revolution. Two different first-order approximations can be applied, where the mechanical speed is estimated by counting the number of pulses in a fixed time (the conventional method in normal speed operation) or counting the time to obtain a fixed number of pulses (the approach that can be considered in very low-speed operation):(22)$$\omega_{m}=\frac{\Delta x}{T}=\frac{x}{\Delta T}$$

In our case, the rotation speed is not too high due to the number of pair poles (*p* = 3), and the conventional approach works fine, providing a good velocity estimation using the 10,000 pulses per revolution that provides the GHM510296R/2500 encoder. The eQEP peripheral is programmed to schedule a fixed-timing interruption event that facilitates the calculation of encoder pulses and the rotor speed between two contiguous events. It can be deduced that the quality of the measurement is proportional to the scheduled time of the programmed interruption, where longer interruption periods result in better (higher-precision) speed measurements at higher sampling times. In our case, different periods have been analyzed, and the eQEP has been programmed with a fixed interruption timing in a steady state of 0.13 ms and 20 ms. This simplifies the rotor speed calculation, providing a good estimation at most operating points. However, some disturbances have been detected when controlling the drive in overshooting operations, and an adaptive algorithm has been implemented in the transient operation of the drive. This algorithm decreases the scheduled time of the programmed interruption, keeping the relative speed error between 0.05% and 1% and reducing the corresponding sampling period of the rotor speed evaluation (eQEP interruption period between 0.05 ms and 5 ms). This means that a quick response is prioritized versus precision in the measurement system during transient drive operation. Figure 9 shows the relative errors obtained in steady state (see the yellow line) and transient (the error is between the blue and red lines) operations. Note that the error axis is plotted in a logarithmic scale, showing a notable increase in its value in low-speed operations.

While the previous methods can be described as the conventional technique for speed measurement due to their widespread use in practice, this paper proposes an alternative algorithm for the evaluation of rotor speed. This technique is based on the ideas of oversampling and noise shaping as in an analog-to-digital converter based on a sigma–delta modulator (SDM) [19]. A rough (inaccurate) speed estimation is made at a high sampling rate, *f_S_* = 20 kHz, so that the errors obtained are shifted to the high-frequency region of the spectrum. The speed thus estimated, *ω_r_*, is then passed through a first-order low-pass filter with a cutoff frequency of 32 Hz to provide strong attenuation of the high-frequency errors. The resulting signal, *ω_f_*, corresponds to a very accurate estimation of the actual speed. Figure 10 shows the functional block diagram of the new algorithm proposed for speed estimation. The outputs provided by the differential encoder are *r_X_*(*t*), where *X* = *A* or *B*. Both signals are square quadrature waves with an instantaneous frequency proportional to the rotor speed, *f_X_*(*t*) = (*N_e_* · *ω_m_*)/(2π), where *N_e_* = 2500 is the number of pulses per revolution and *ω_m_* is the mechanical speed. The peripheral eQEP generates a sequence of pulses placed at the times where the edges of *r_A_*(*t*) and *r_B_*(*t*) are located. Consequently, the asynchronous counter increases at the rate of *R_C_*(*t*) = 4 · *f_X_*(*t*) = (4 · *N_e_ · ω_m_*)/(2π) units per second. Finally, the output of the counter is sampled at the rate *f_S_* and differentiated to obtain the speed estimation. It is shown in [20,21,22] that
(23)$$\omega_{r}\left(n\right)=\frac{R_{c}\left(t\right)}{f_{s}}+\left[q\left(n\right)-q\left(n-1\right)\right]=\frac{4N_{e}\omega_{m}}{2\pi f_{s}}+\left[q\left(n\right)-q\left(n-1\right)\right]$$
where *q*(*n*) is the quantization error due to the time difference between the pulses at the eQEP output, *r*(*t*), and the sampling instants, *f_S_*, which is expressed in the z-domain as
(24)$$\omega_{r}\left(z\right)=\frac{4N_{e}}{2\pi f_{s}}\cdot \omega_{m}\left(z\right)+\left(1-z^{-1}\right)\cdot Q_{z}$$

Note that the quantization error *Q*(*z*) is filtered using the first-order high-pass transfer function *F*(*z*) = (1 − *z*^−1^); that is, the error is very small at low frequencies (the gain is *F*(1) = 0 at DC) and takes larger values at high frequencies (the maximum is reached at *f* = *f_s_*/2, where the gain is *F*(–1) = 2) [19]. For illustration purposes, the architecture in Figure 10 was simulated for inputs *r_A_*(*t*) and *r_B_*(*t*) corresponding to a rotor speed *ω_m_*(*t*) with an offset (*ω_off_*) of 70 rad/s and an AC component of a frequency of 10 Hz and an amplitude (*ω_pk_*) of 65 rad/s. These values were chosen to achieve a variation from low to high speeds. The density of the power spectrum of *ω_r_*(*n*) is shown in Figure 11a. Two spectral peaks appear, corresponding to the DC and the 10 Hz AC components. As expected, the quantization error components present the characteristic noise shaping of the SDM output; the noise background is very small at low frequencies and increases at a rate of 20 dB/dec (first-order performance).

The estimation can then be filtered to attenuate the high-frequency components, and a simple first-order low-pass filter with a cutoff frequency of 32 Hz was implemented, which was approximated with the following transfer function in the z domain:(25)$$H\left(z\right)=\frac{2\pi f_{s}}{4N_{e}}\cdot \frac{0.01}{1-z^{-1}}$$

As can be observed in Figure 11b, the filtered signal, *ω_f_* (*n*), and the rotor speed, *ω_m_*(*t*), match very well, and the difference is very slight. The noise component obtained in the simulation is 0.0013, and the power of the filtered signal components can be calculated as follows:(26)$$P_{\omega_{f},s}=\omega_{off}^{2}+\frac{\omega_{pk}^{2}}{2}=7013$$

In conclusion, for the considered rotor speed and for the ideal implementation of the proposal, a signal-to-noise ratio (SNR) of 67.2 dB can be achieved.

Figure 12 shows the relative errors obtained, showing a better performance than using the conventional method. Note that a filter or an order higher than one can be easily applied, notably reducing the relative errors. Figure 13 compares the performance of the rotor speed evaluation using the analyzed methods and the experimental test rig. The upper plots (left and right sides) show the performance using the conventional method with a fixed programmed eQEP interrupt timing of 0.13 ms (left side) and 20 ms (right side). The lower plots depict the behavior using the conventional algorithm with the adaptive algorithm (left side) and the oversampling measurement technique (right side).

## 4. Verification Analysis Using the Experimental System

The calibration curves obtained are experimentally tested using the experimental system during normal operation, where steady-state and transient regimes are considered. In our case, the multiphase drive is regulated using an outer speed controller that generates the stator current references for an inner stator current regulator based on the predictive control technique, a finite control set model predictive controller or FCSMPC, as stated in Figure 14. Further details of the considered control system can be found in [18]. The conventional measurement method is used for stator currents, whereas the oversampling technique is used for speed measurement. Different steady and transient state trials, including a reversal test, are performed to observe the performance of the stator currents and the rotor speed, where different load torques were applied. Figure 15, Figure 16, Figure 17 and Figure 18 summarize the results obtained, where good tracking of the controlled variables can easily be appreciated in all the cases analyzed.

## 5. Conclusions

Although the first electrical drive was built in the 19th century, the topic of electrical drives has notably grown in recent times, when the industrial and the scientific communities have been demanding and looking forward to high-performance drives, which are popular targets of the efficiency, robustness, preciseness, and fault-tolerant capability control of the system. These targets require good and well-known monitoring of the electrical drive, which depends on the calibration process of the stator current and rotor speed sensors. Our work focuses on this calibration process, where different methods are illustrated, compared, and validated in a laboratory setup using a particular multiphase drive based on a five-phase induction machine. This analysis can be of interest to practitioners in the field of electric drives that can use the described measurement techniques in future product developments.

## Figures and Tables

**Figure 1 sensors-23-07317-f001:**
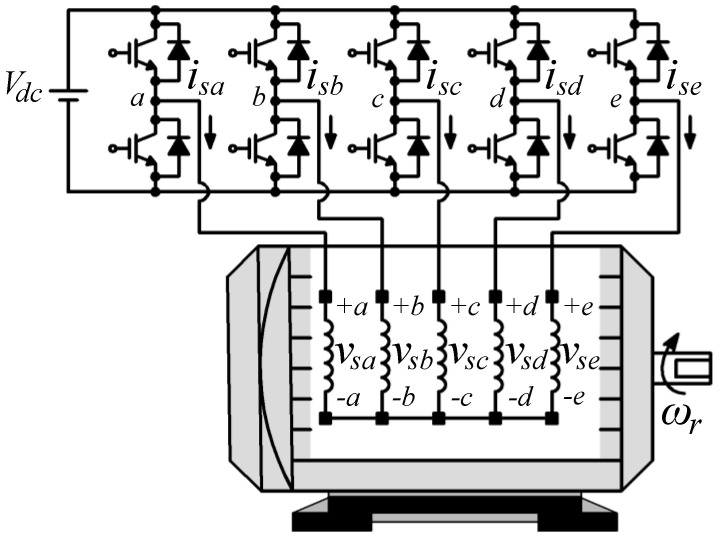
Schematic diagram of the system analyzed.

**Figure 2 sensors-23-07317-f002:**
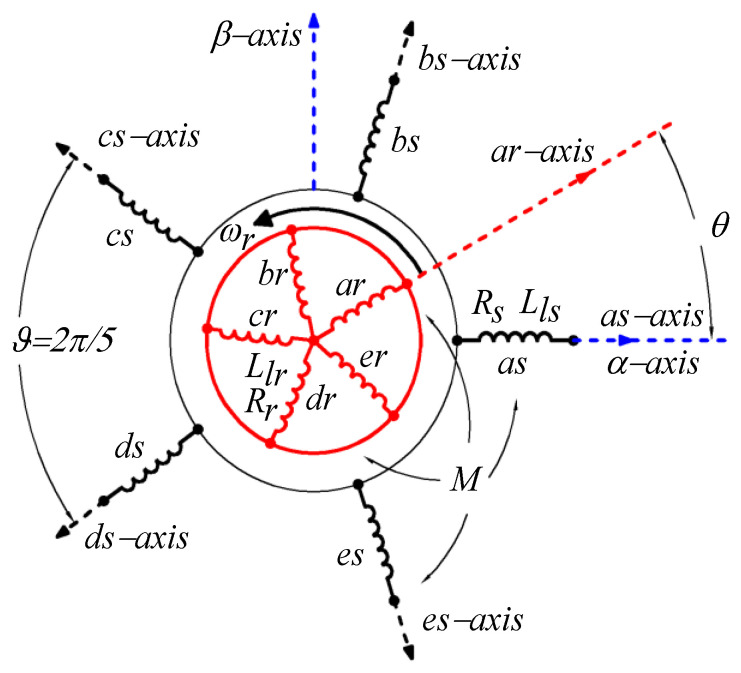
Five-phase squirrel-cage induction machine.

**Figure 3 sensors-23-07317-f003:**
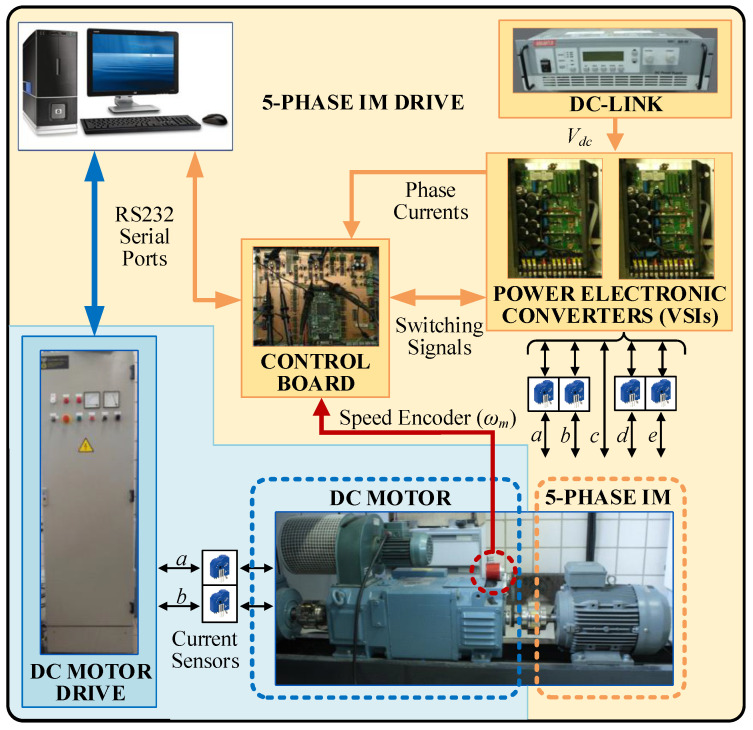
Experimental test rig based on a multiphase drive.

**Figure 4 sensors-23-07317-f004:**
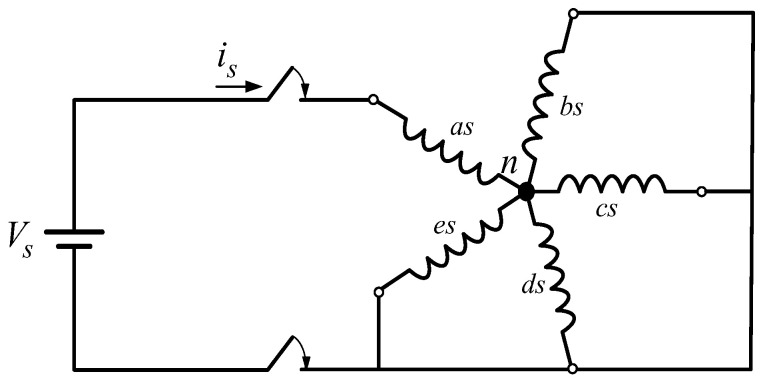
Calibration test.

**Figure 5 sensors-23-07317-f005:**
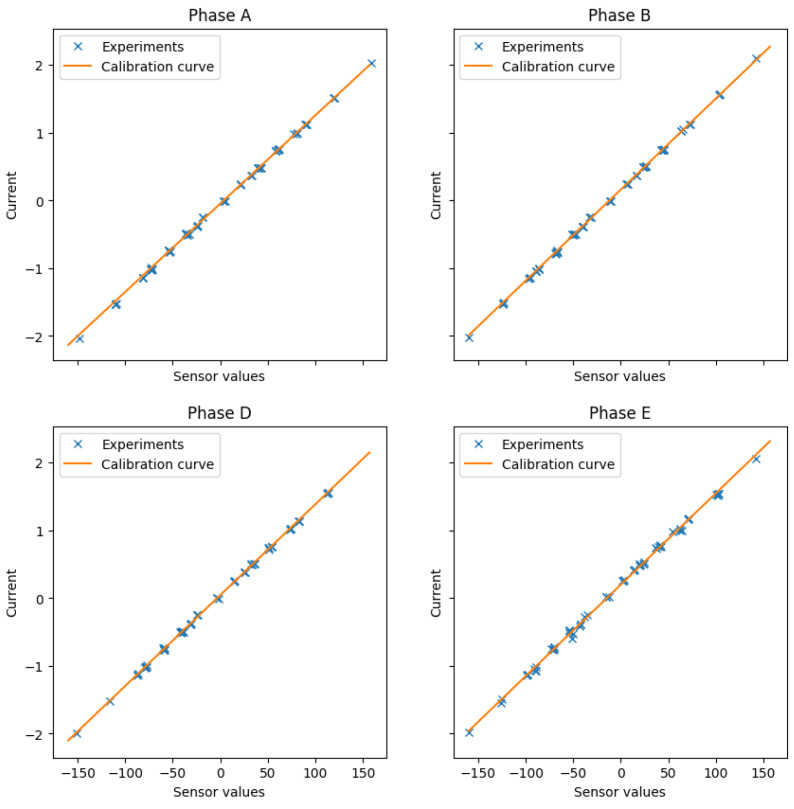
Results obtained using the conventional calibration test.

**Figure 6 sensors-23-07317-f006:**
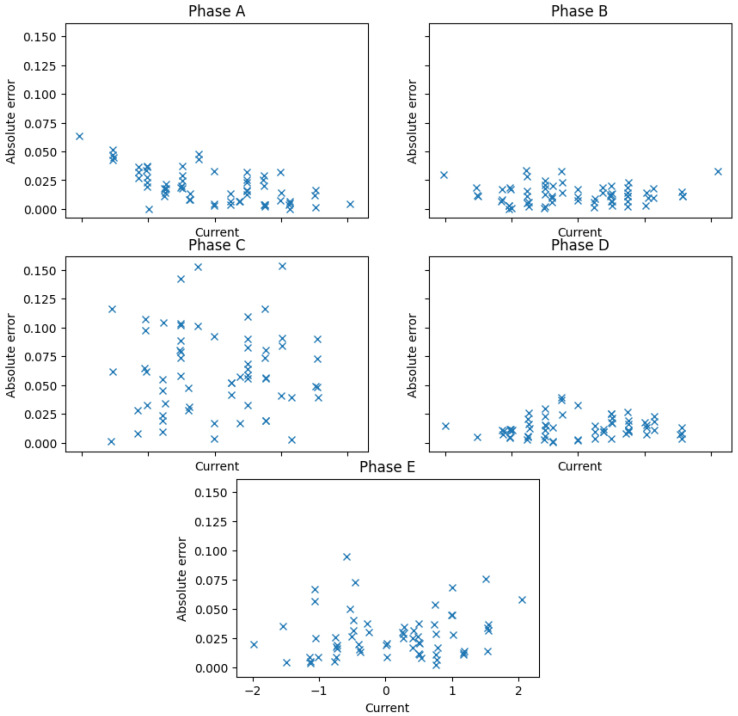
Absolute errors after the conventional calibration test.

**Figure 7 sensors-23-07317-f007:**
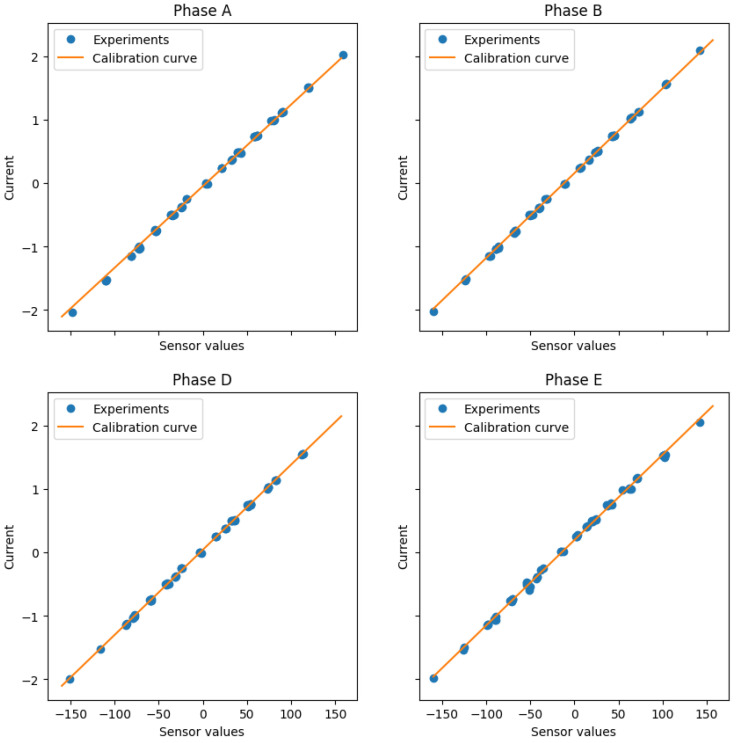
Results obtained using the generalized calibration test.

**Figure 8 sensors-23-07317-f008:**
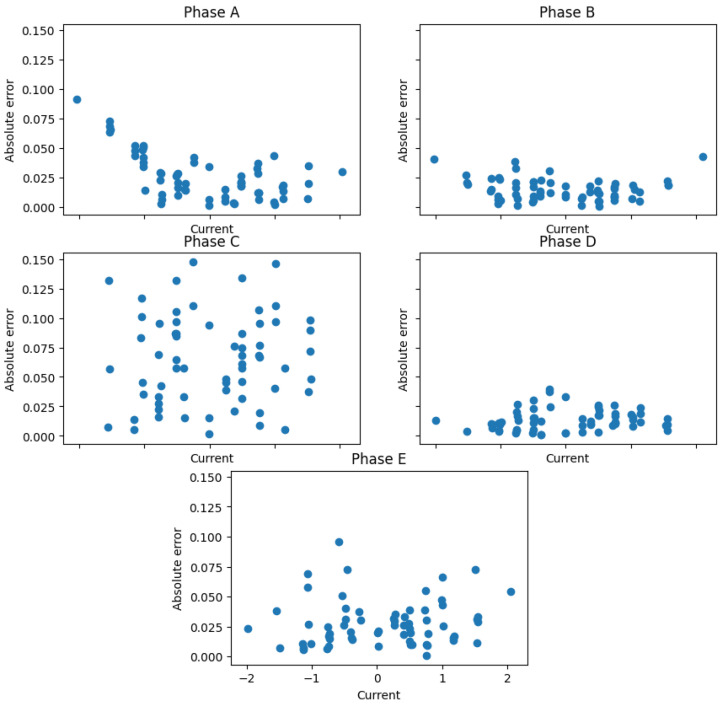
Absolute errors after the generalized calibration test.

**Figure 9 sensors-23-07317-f009:**
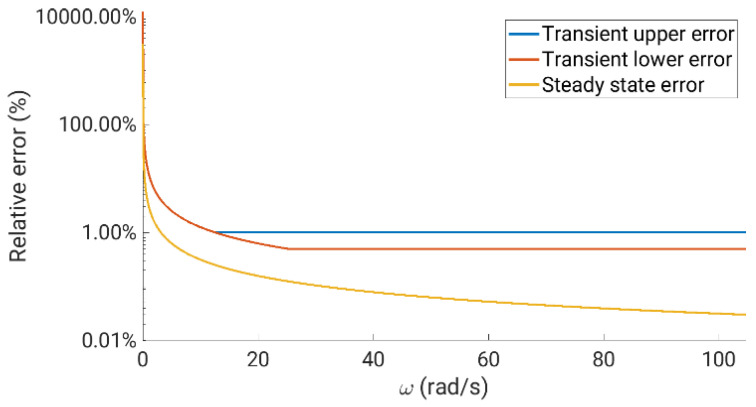
Relative error rate obtained in the speed range using the conventional techniques.

**Figure 10 sensors-23-07317-f010:**
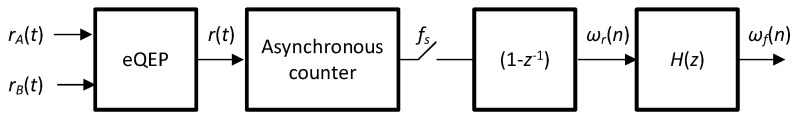
Laboratory setup in the verification analysis.

**Figure 11 sensors-23-07317-f011:**
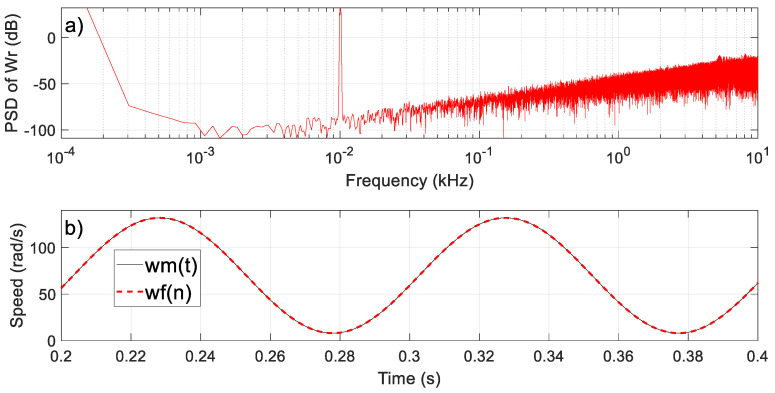
Performance of the proposal for a particular speed estimation using simulation results and a Matlab/Simulink^®^ environment. The upper plot shows the density of the power spectrum of *ω_r_*(*n*), while the lower plot shows the accuracy of the estimation, *ω_f_*(*n*).

**Figure 12 sensors-23-07317-f012:**
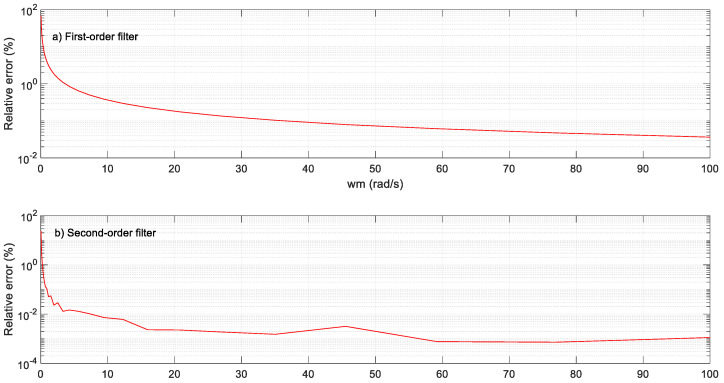
Relative error rate obtained in the speed range using the oversampling method and a first-order filter (upper plot) and a second-order filter (lower plot).

**Figure 13 sensors-23-07317-f013:**
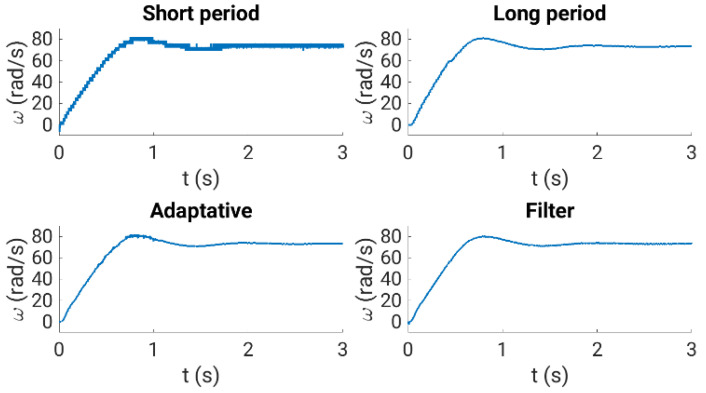
Performance of the rotor speed measurement using different methods. Upper plots: performance using the eQEP and the conventional method in normal-speed operation with a short time period (**left side**) or a long time period (**right side**). Lower plots: performance using the eQEP and the adaptive proposal (**left side**) and the oversampling technique (**right side**).

**Figure 14 sensors-23-07317-f014:**
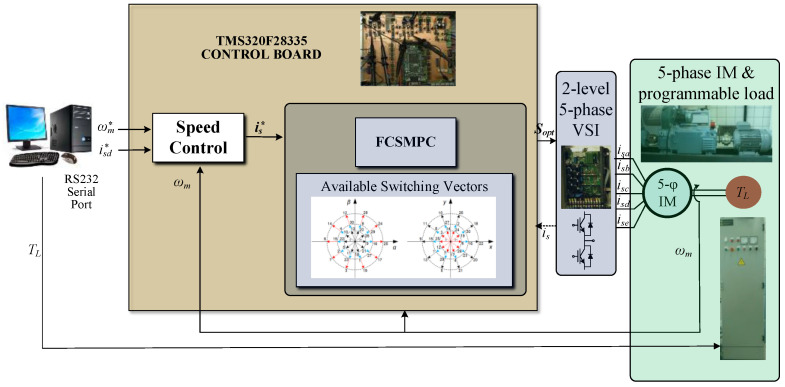
Laboratory setup in verification analysis.

**Figure 15 sensors-23-07317-f015:**
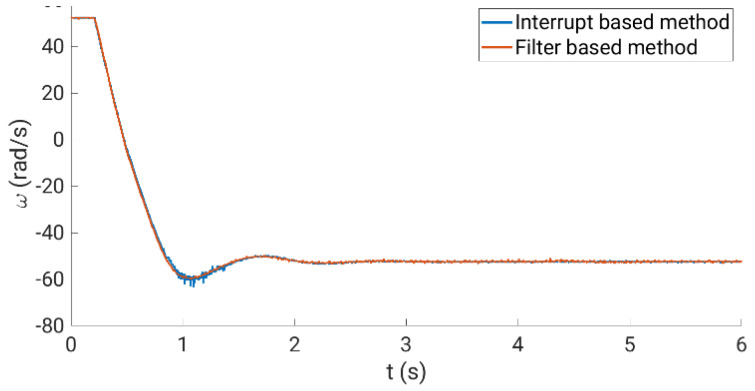
Rotor speed obtained (blue line) following the reference signal (red line).

**Figure 16 sensors-23-07317-f016:**
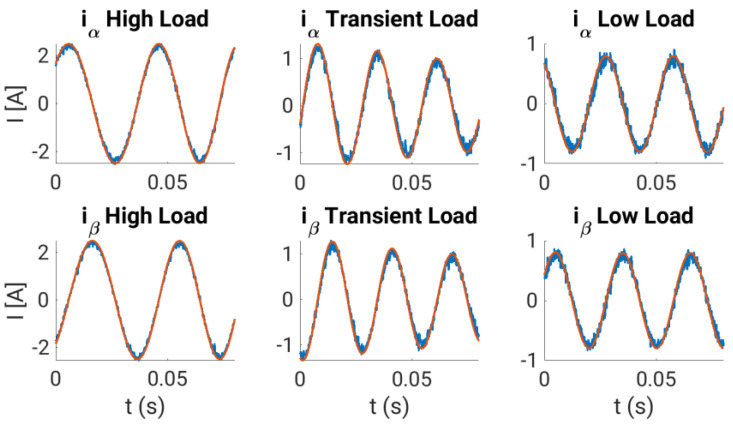
Stator current measured (blue traces) following reference signals (red traces) in the verification analysis. Different load torques are applied during the experiments. For simplicity, one stator current is considered (note that similar performances are obtained in all the phases).

**Figure 17 sensors-23-07317-f017:**
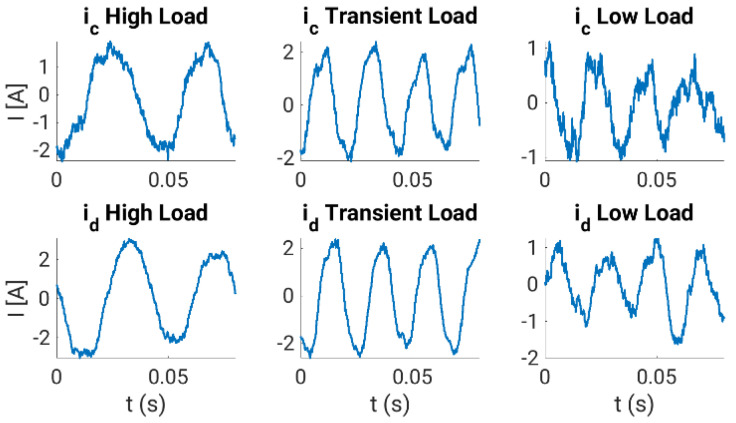
Performance of the unmeasured (upper plots, phase *d*) and one measured (lower plots, phase *c*) stator currents in different situations.

**Figure 18 sensors-23-07317-f018:**
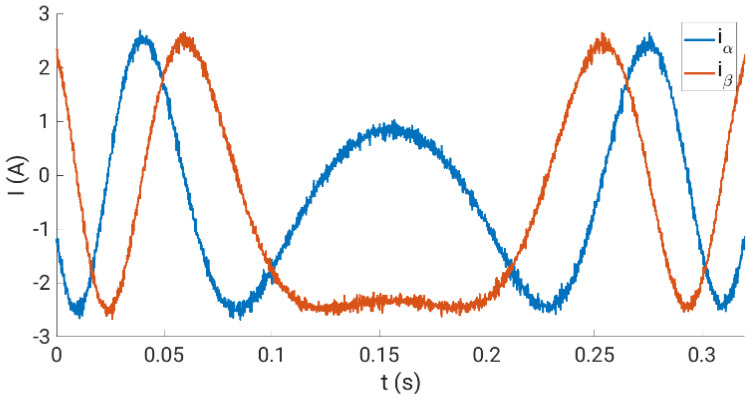
Stator currents in the *α*-*β* plane at the zero-speed crossing point during the reversal test.

**Table 1 sensors-23-07317-t001:** Data of the induction machine used in the experiments.

Source	Parameter	Value/Description
	Rated current (Arms)	7.13
Original nameplate data	Rated power (kW)	4
	Rated speed (r/min)	2880
	Conductor	Copper
	Diameter (mm)	0.7
	Number of pole pairs	3
Rewinding data	Number of slots	30
	Number of turns	165
	Type of windings	Single-layer
	Winding pitch	5 slots (full pitch)
	Stator resistance, *R_s_*	12.85 Ω
	Rotor resistance, *R_r_*	4.80 Ω
Electrical parameters	Stator leakage inductance, *L_ls_*	79.93 mH
	Rotor leakage inductance, *L_lr_*	79.93 mH
	Mutual inductance, *M*	681.7 mH
Mechanical parameter	Rotational inertia, *J_m_*	0.02 kg-m^2^

## Data Availability

The data used to support the findings of this study are available from the corresponding author upon request.

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
