# Peer review of "On-Site Calibration of an Electric Drive: A Case Study Using a Multiphase System"

_sensors, 2023, doi:10.3390/s23177317_

Round 1

Reviewer 1 Report

1. The work focuses on the calibration process of a particular multiphase drive based on a five-phase induction machine, where different methods are presented, compared, and validated in a laboratory set-up.

2. This study examines various calibration methods for current and speed sensors, with a specific focus on an example case involving a five-phase induction motor drive.

3. The statement: „number of sensors is usually minimized to reduce the total cost of the system”, is not completely right. It must be substantiated or supported by bibliographic references. See the introduction section.

4. A current State of the Art should expose something that performs in the field, not something that is already known: „ Electric drives play an important role in modern societies, being the mainstay of many industrial applications such as electric vehicles, elevators, and robotics [1]. They are systems based on electric machines and controlled with power electronic converters, whose goal is accurate transference between electrical energy and mechanical movement with fast dynamic responses [2]. Among modern electric drives, those including multiphase machines has become a reality in recent decades [3] „

5. The first part of the introduction section should be reconsidered. Research in the field should be concisely presented. The bibliographical references are questionable and do not sufficiently expose recent research.

6. The topic presents some degree of novelty, and it is original and relevant to the field. It addresses a specific gap in the field of feedback used in controlling the state of five-phase induction motor drive.

7. The work has relevant experimental research, and it demonstrates and validates the initial statements presented in the introduction section.

8. The mathematical part is well founded, and it is clearly experimentally validated.

The English language should be checked. Parts of the paper must be checked before submission!

Author Response

Reviewer #1

The authors thank the reviewer for the provided comments. We feel that, by incorporating the suggestions, the paper has improved in terms of quality and clarity.

  1. The work focuses on the calibration process of a particular multiphase drive based on a five-phase induction machine, where different methods are presented, compared, and validated in a laboratory set-up.
  2. This study examines various calibration methods for current and speed sensors, with a specific focus on an example case involving a five-phase induction motor drive.
  3. The statement: „number of sensors is usually minimized to reduce the total cost of the system”, is not completely right. It must be substantiated or supported by bibliographic references. See the introduction section.

Please note that some new references have been included in the revised manuscript to substantiate the statement. Thanks for the appreciation.

  1. A current State of the Art should expose something that performs in the field, not something that is already known: „ Electric drives play an important role in modern societies, being the mainstay of many industrial applications such as electric vehicles, elevators, and robotics [1]. They are systems based on electric machines and controlled with power electronic converters, whose goal is accurate transference between electrical energy and mechanical movement with fast dynamic responses [2]. Among modern electric drives, those including multiphase machines has become a reality in recent decades [3] „
  2. The first part of the introduction section should be reconsidered. Research in the field should be concisely presented. The bibliographical references are questionable and do not sufficiently expose recent research.

Please note that the paragraph has been rewritten, and new references have been included in relation with the state of the art of the provided research (the new version of the work has 22 references in total).

  1. The topic presents some degree of novelty, and it is original and relevant to the field. It addresses a specific gap in the field of feedback used in controlling the state of five-phase induction motor drive.
  2. The work has relevant experimental research, and it demonstrates and validates the initial statements presented in the introduction section.
  3. The mathematical part is well founded, and it is clearly experimentally validated.

Again, the authors would like to thank the reviewer for the provided comments about their work.

Reviewer 2 Report

The main question addressed by the research is the analysis of different calibration techniques of current and speed sensors. Since there is little scientific evidence of the influence of the calibration technique in the measurement of the stator currents, the authors have analyzed these influences, establishing a reliable procedure for the calibration of the analyzed system that can be extrapolated to any type of electrical drive controlled in speed and current.

Therefore, I consider this research being very important for the field.

There is an error, probably caused by word processing in line 192: “Error! Reference source not found.”
I assume this must be corrected before acceptance. There are a few other ones of the same type in lines 205-206, 213, 259-260. This is weird because there are several good (indexed) references in the Introduction chapter.
If it is possible for the Publisher’s technical editors to surely identify the relevant references, the paper can be accepted “as is”. Otherwise, it must be returned to the authors to check and identify the references.

Author Response

Reviewer #2

The main question addressed by the research is the analysis of different calibration techniques of current and speed sensors. Since there is little scientific evidence of the influence of the calibration technique in the measurement of the stator currents, the authors have analyzed these influences, establishing a reliable procedure for the calibration of the analyzed system that can be extrapolated to any type of electrical drive controlled in speed and current.

Therefore, I consider this research being very important for the field.

The authors thank the reviewer for the comments and for time and effort dedicated to the review. We feel that, by incorporating the suggestions, the paper has improved in terms of quality and clarity.

There is an error, probably caused by word processing in line 192: “Error! Reference source not found.” I assume this must be corrected before acceptance. There are a few other ones of the same type in lines 205-206, 213, 259-260. This is weird because there are several good (indexed) references in the Introduction chapter. If it is possible for the Publisher’s technical editors to surely identify the relevant references, the paper can be accepted “as is”. Otherwise, it must be returned to the authors to check and identify the references.

The reviewer is right, and the typos have been solved. Thanks for raising the issue.

Reviewer 3 Report

Thank you for your contribution to “Sensor”. I recognized that this paper is about the calibration of a 5-phase induction motor drive for cost reduction in controlling electrical equipment.

Although the contents are summarized briefly, at least 20 to 30 papers should be reviewed in order to show the originality and new knowledge of this research as an international journal. I doubt the review will be limited to 7 papers in the introduction.

There are many reference errors in line 192, 205-206, 213, and 259-260.

Author Response

Reviewer #3

Thank you for your contribution to “Sensor”. I recognized that this paper is about the calibration of a 5-phase induction motor drive for cost reduction in controlling electrical equipment.

Although the contents are summarized briefly, at least 20 to 30 papers should be reviewed in order to show the originality and new knowledge of this research as an international journal. I doubt the review will be limited to 7 papers in the introduction.

The authors thank the reviewer for the comments and for time and effort dedicated to the review. We feel that, by incorporating the suggestions, the paper has improved in terms of quality and clarity. The number of references in the introduction section has been updated, and the submitted manuscript has a total number of 22 references.

There are many reference errors in line 192, 205-206, 213, and 259-260.

The reviewer is right, and the typos have been solved. Thanks for raising the issue.

Round 2

Reviewer 3 Report

Dear Authors,

My minimal remarks have been appropriately corrected.

Author Response

We appreciate your work reviewing the different versions of the manuscript. Thanks for your valuable contribution to making the submitted document much more interesting for the scientific community.